# Households' Willingness to Accept Forest Conservation and Ecosystem Services

## Meiyan Zhang

School of Economics and Management, Low Carbon Economy Research Center of Humanities and Social Sciences Research Base of the Universities in Fujian, Sanming University, Sanming 365004, China; 20171209@fjsmu.edu.cn

**Abstract:** In this study, forest owners' willingness to accept the governmental redemption of commercial forests for forest conservation, as well as the factors influencing their willingness, was analyzed. It was found that having expected non-timber income from conservation programs, trustable government policies, simpler dealing with government departments for disputes, and satisfactory local ecological condition had strong impacts on the likelihood of participation for the households. If the sum of direct cash compensation incentives and indirect non-timber income compensation incentives was greater than the opportunity costs incurred by forest owners for protection, forest owners were more willing to participate in the redemption. Based on the results, the final offer arbitration method was recommended to improve the maximum price method for redemption, which enables forest owners to receive recognized incentives for direct cash compensation. Ecotourism was strongly recommended to raise forest owners' expectations of sustainable non-timber income and deliver on such expectations with lower information costs.

**Keywords:** forest conservation easement; economic incentive; forest governance; Fujian

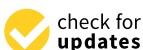



## 1. Introduction

As a key terrestrial ecosystem, forests provide wood and other forest products, in addition to playing an irreplaceable role in climate change mitigation, water regulation, soil conservation, and biodiversity. Changes in economic development, social needs, and forestry policies affect the transformation of forest resource utilization [1–3]. In the early stage of economic development, the social demand for forests is mainly wood. When the economic development reaches a certain stage, the social demand expands from wood to ecological services, and the ecological value of forests would continues to increase along with economic development.

Changing relative scarcity of forest products or ecological services calls would for changes in forestry policies [4]. When a more ecological value is given to forestland due to factors such as water conservation, aesthetic landscape, and biodiversity, land value for wood production relatively decreases, and logging restrictions and logging ban policies may need to be imposed on initially classified commercial forests. To change land use, regulation and compensation mechanisms should be introduced to realize the ecological value of forests [5]. To increase the effective supply of forest ecological services, the state has launched various forest protection projects, a typical representative of which is the redemption of commercial forests in key ecological locations in China.

China's collective forests were distributed to households as commercial forests several years ago when ecological services and conservation had not been fully considered. To engage the commercial forests allocated to households for conservation purposes, which is increasingly important, the government is adopting redemption of the rights for conservation purposes. Redemption is like conservation easement practiced in many other countries [6]. It usually starts in pilot counties (cities) in China. During the start-up phase, a leading organization is established to actively raise funds and conduct supervision, assessment, and evaluation, and carry out extensive and continuous mobilization.

When a county (city) is considered, the distribution of commercial forests is investigated, and mapping is conducted before formulating the annual redemption plan and public announcements. After considering the interested forest owners, the committee reviews and assesses the forestland, and shortlisted owners are identified for bidding as candidates for final assessment. If the owner has no objection to the assessment report, a redemption contract can be signed and announced. After the announcement, the forest allocation, forest ownership change, and fund payment can be carried out. After the redemption is completed, the governing committee manages the forest following ecological priority and gradually allocate the redeemed commercial forests into ecological forests providing ecosystem services other than timber production. There are 651,600 hectares of forests categorized as commercial forests in key ecological locations in Fujian Province. Twenty-three provincial-level reform pilots were determined and scheduled in three phases from 2016 to 2018. It was proposed to redeem 13,333 hectares from 2016 to 2020. As of November 2020, the redemption of 22,400 hectares had been completed ahead of schedule.

To better understand the redemption progress, we selected reforms at five pilot projects in Yong'an City, Shaxian District, Jiangle County, Ninghua County, and Jianning County, to conduct field investigations. The specific redemption situation is shown in Table 1. As of August 2020, Yong'an City has raised a total of 119 million yuan, completed the redemption of 2933 hectares, and implemented forest quality improvement of 493.33 hectares. From 2017 to July 2020, Shaxian District invested a total of about 18 million yuan, part of which was used to redeem natural commodity forests, completing the redemption area of 416.8 hectares. Jiangle County implemented the PPP project for the protection of forest resources in key ecological locations to promote redemption. Jinshan Forest Farm Co., Ltd., Sanming, raised 130 million yuan in its own capital and applied for 400 million yuan in project financing from the China Development Bank.

**Table 1.** Redemption status of commercial forests in key ecological locations in the study area (unit: ha).

| | Implementation Time | Area of Commercial Forests | Cumulative Redemption Area | Redemption Entity |
|---|---|---|---|---|
| Yong'an City | 2013 | 14,000.00 | 2933.33 | Yong'an Ecological Civilization Volunteer Association, Sanming, China |
| Shaxian District | 2015 | 11,600.00 | 416.80 | Shaxian Forest Resources Purchase and Storage Management Co., Ltd., Sanming, China |
| Jiangle County | 2017 | 7680.00 | 4000.00 | Jiangle County Jinshan Forest Farm Co., Ltd.,Sanming, China |
| Ninghua County | 2018 | 25,166.67 | 614.73 | Ninghua County State-owned Ecological Forest Farm Co., Ltd., Sanming, China |
| Jianning County | 2018 | 24,133.33 | 906.67 | Jianning County Minjiangyuan State-owned Forest Farm Co., Ltd., Sanming, ChinaJianning County Forestry Construction Investment Company, Sanming, China |

The deadline for statistics in the 5 counties and cities was the survey day.

Ninghua County's State-owned Ecological Forest Farm Co., Ltd., Sanming, redeemed 253 hectares in 2018 and 361 hectares in 2019. The total redemption expenditure was 8.87 million yuan in 2019, of which the provincial government allocated 5.4 million yuan, and the gap was financed by the county. From 2018 to June 2020, Jianning County raised various redemption funds of 43.75 million yuan, comprising the central key ecological protection and restoration special fund of 8.2 million yuan and various provincial supporting funds with a total of 35.37 million yuan; the completed redemption area was 907 hectares.

## 2. Literature Review

Voluntary-based conservation programs have received increasing attention, and understanding forest owners' participation behavior in such programs has become a crucial part of policy success [7–9]. A growing number of studies examine forest owners' participation in conservation programs considering owners' characteristics, objectives, forest conditions, and policy variables [7,10,11]. Mitani and Lindhjem (2015) found the owners' expectation of sustainable non-timber income from reserve-related commercial activities to be over and beyond the compensation payment itself, which drives sufficient motivation to participate in forest conservation incentive programs [12]. Economic incentives might change the behavior of the owners in general, but their efficacy may vary for different owners. For example, owners whose main objective is not timber production may be willing to accept lower compensation. Thus, the effectiveness of incentive programs may be a context-dependent, empirical question [13].

In addition to economic incentives, public governance is also another important factor that affects the implementation of the policy. According to the governance system in Ostrom's social–ecological system analysis framework, the system-related governance is included in the analysis framework, which encourages forest farmers to voluntarily participate in commercial forest redemption projects [14]. Therefore, an important factor for better implementation of the policy is that the government establishes a mutually beneficial positive incentive relationship between stakeholders in the incentive mechanism [15].

In this paper, a survey was conducted of representative forest owners to investigate their willingness to accept (WTA) in participating in the government acquisition of commercial forests for conservation in Fujian, China. The variables chosen allowed an examination of the impacts of economic incentives, government governance, land and resource conditions, and ownership and family characteristics. In this article, we sought to explore the core factors to cope with the conflict between logging and economic use and conservation of the forests under the governmental budget.

## 3. Theoretical Framework

Providing incentives for economic entities is essential to induce owners to voluntarily participate in the redemption. Lowest transaction costs (mainly the information cost) and incentive compatibility with other goals such as logging restrictions and logging ban policies, to achieve Pareto improvement, have transformed commercial forests in key ecological locations from providing private goods to mixed products. The positive externalities from regulation and compensation mechanisms should be the base for internalization of externalities so as to realize the ecological value of forests in the process of maximiing economic value of forests [5].

It is assumed that the government and forest owners seek to maximize their own interests. The government's primary goal is to have ecological conservation. In contrast, forest farmers' goal is to increase private income. When the direct income of forest owners from redemption is insufficient to compensate for their losses, a continuous non-timber income is required to supplement the loss to ensures the coordination of personal interests with the overall interests of society.

The expected compensation would largely determine whether they participate in a commercial forest redemption project. The direct cash compensation standard that forest owners can obtain is set uniformly by the local government. For example, Yong'an City has a total of 14,000 $hm^2$ of commercial forests in key ecological locations. Around 210–320 million yuan from public finances is required to redeem the forests if we use the current redemption price, and the standard value for artificial commercial forests is 22,500–33,750 $yuan/hm^2$; for natural forests, it is 7500 $yuan/hm^2$. However, the government can only allocate 15 million yuan for redemption every year. Thus, the redemption standard unilaterally set by the government is lower than the real value of commercial forests in key ecological locations. If the market price of stumpage is used, its value is about 75,000–120,000 $yuan/hm^2$ [5].

If the government adopts a simple and easy-to-operate, one-time direct compensation method in the absence of relevant measures and supporting policies, the program is unlikely to be successful. The owners are unlikely to continue long-term investment and management of the ecosystem service if their income does not meet their expected goals [16]. The opportunity cost for the farmers enrolled in the program is the loss of opportunity income that cannot be sold through the logging of commercial forests, and the income is the direct cash compensation provided by the government and the non-timber income earned by forest owners. Therefore, the likelihood of forest owners participating in the redemption can be expressed as

$$E_f = F(D + I - WTA - C_1) \tag{1}$$

where $E_f$ is the likelihood of participation, D represents the government's direct cash compensation, I is the indirect non-timber income compensation incentives received by forest owners, WTA is the opportunity value the owner can have if not engaged in the redemption, and $C_1$ is the transaction cost of forest owners choosing to participate in the redemption policy, including information search costs, forest resource asset evaluation costs, negotiation costs, contracting, and implementation costs.

If cash compensation incentives and indirect non-timber income are greater than the opportunity loss from not logging the forests and transaction costs, the farmers are more likely to participate in the redemption. Higher D and I values and lower WTA and $C_1$ values increase the likelihood of participating in redemption.

The value of the conservation can be expressed by WTP, which represents the government's maximum willingness to pay. Then, the government's likelihood for redemption can be expressed as

$$E_g = G(WTP - D - C_2) \tag{2}$$

where $E_g$ represents the likelihood of the government to redeem, and $C_2$ is the transaction cost for the government, including identifying and managing the program other than direct payment.

Equations (1) and (2) show that incentive compatibility is finally achieved through incentives and coordination among stakeholders of ecological protection [17]. When the indirect incentives for non-timber income increase, direct cash compensation can be reduced, and the financial pressure is eased. In the transformation of the leading forestry industry, whether forest owners can obtain sustainable non-timber income is the key to transforming their traditional livelihood model into a sustainable livelihood model.

## 4. Data and Method

The data were collected from a field survey conducted by the research team in Yong'an City, Sha County, Jiangle County, Ninghua County, and Jianning County in Fujian Province from July to August 2020. The survey was carried out by group discussions within forestry bureaus, state-owned forest farms, and other redemption entities and village committees. Large-scale forest owners participating in the redemption were selected to conduct semi-structured interviews, mainly focusing on understanding redemption behavior, awareness, evaluation, and impact. Household interviews were conducted to investigate small-scale forest owners according to a standardized questionnaire. Based on the information obtained in group discussions, the research team first found the villages that had a deep understanding of the redemption policy, used the house number as the number of each household, and drew samples according to the principle of simple random sampling to ensure that the sample size of each village was basically the same. The questionnaire was developed consisting of four main sections: (1) background information about the forest owners and their households, including age, education, income, membership in government offices, and the number of old people; (2) the plot condition and the forest property; (3) economic incentives factors, including inquiring about the income levels of forest owners from various sources in the previous year, and more importantly, using the Likert scale to measure the importance of various income-generating activities;

(4) government governance factors, including the relationship between the government and forest owners, the efficiency of government administration, and forest owners' attitudes toward the government's governance of the ecological environment. A survey of a total of 240 households was conducted, generating 218 valid questionnaires, with a valid response rate of 90.83%.

Tables 2 and 3 report the variables used for the empirical analysis, as well as their definition and mean and standard deviation. The dependent variable (Willingness) was the forest owner's willingness to participate in the redemption. Among the sample of 218 respondents who answered the willingness question, 79% were willing to participate. Based on the theoretical framework, four types of relevant explanatory variables were determined, which are shown in Table 2.

**Table 2.** Definition of explanatory variables.

| Variable | Description |
|---|---|
| DV: Dependent Variable | |
| Willingness | Dummy: Owners' willingness to participate in the redemption |
| OH: Owner/Household Characteristics | |
| Servant | Dummy: There are government servants in the owner's household |
| Age | Age of owner |
| Edu | Education of owner (1: below elementary school; 2: elementary school; 3: junior high school; 4: high school; 5: junior college; 6: university) |
| Income | Annual income of owner (ten thousand yuan) |
| Number | Number of seniors over 60 years old in the owner's household |
| LR: Land/Resource Conditions | |
| Size | Size of forest (1/15 ha) |
| Mature | Percentage of mature forest |
| EI: Economic Incentives | |
| Timsale | Economic importance of timber sale (1: not important at all; 5 = very important) |
| Nontimincome | Expectation of sustainable non-timber income from conservation programs(1: strongly disagree; 5: extremely agree) |
| GG: Government Governance | |
| Trust | Trust in government policies (1: strongly doubt; 5: very confident) |
| Frequency | Frequency of contacts with forestry bureaus and other government departments due to disputes (1: never; 2: occasionally; 3: often) |
| Sat1 | Satisfaction with local ecology (1: very dissatisfied; 5: very satisfied) |
| Sat2 | Satisfaction with ecological compensation of public welfare forest (same as above) |

The characteristics of the owners and their households (OH) included whether there were government servants in the owner's household (Servant), age (*Age*), education (Edu), income (Income), and the number of seniors over 60 years old (Number). The government servants accounted for 7.9% of respondents, the average age was 48.73, the average level of education was junior high school, the average annual income was 55,100 yuan, and the average number of seniors was 1.61. For comparison, as shown in Table 3, the older the owner, the higher the education, and the more the number of the elderly, the less willing to participate in the redemption of commercial forests. Higher personal annual income and a higher proportion of servants led to an increase in willingness to participate in the redemption.

**Table 3.** Descriptive statistics of different types of forest owners.

| | Mean of Willing to Participate (Standard Deviation) | Mean of Unwilling to Participate (Standard Deviation) | All Sample Mean (Standard Deviation) |
|---|---|---|---|
| Number of samples | 173 | 45 | 218 |
| Proportion (%) | 79 | 21 | 100 |
| Servant | 8.00 (0.27) | 7.70 (0.27) | 7.90 (0.27) |
| Age | 48.49 (10.31) | 49.65 (9.17) | 48.73 (10.06) |
| Edu | 3.04 (1.15) | 3.08 (1.06) | 3.05 (1.13) |
| Income | 5.87 (6.49) | 4.12 (2.91) | 5.51 (5.97) |
| Number | 1.57 (0.91) | 1.77 (0.99) | 1.61 (0.93) |
| Size | 19.12 (26.23) | 15.12 (15.40) | 18.29 (24.38) |
| Mature | 66.51 (28.13) | 51.56 (33.29) | 63.52 (29.70) |
| Timsale | 3.74 (1.11) | 4.35 (0.98) | 3.87 (1.11) |
| Nontimincome | 3.70 (0.82) | 3.12 (1.03) | 3.58 (0.90) |
| Trust | 4.02 (0.65) | 3.31 (0.88) | 3.87 (0.76) |
| Frequency | 1.33 (0.47) | 1.39 (0.50) | 1.34 (0.48) |
| Sat1 | 3.68 (0.98) | 3.42 (0.81) | 3.63 (0.95) |
| Sat2 | 2.77 (0.90) | 2.62 (0.70) | 2.74 (0.86) |

The variables of owners' land or resource conditions (LR) included the total size of their forest (Size) and the estimated proportion of mature forest (Mature). The average total size was about 1.22 hectares, and on average, 63.52% of owners' forests were reported as mature forests. For comparison, the larger the size was, and the higher the proportion of a mature forest, the more willing the forest owner was to participate in the redemption.

The economic incentive variable (EI) included the owners' consideration of the economic importance of income from timber (Timsale) and their perceived expectation of sustainable non-timber income from conservation programs participation (Nontimincome). Two five-scale categorical variables (Timsale, Nontimincome) indicated the owners' attitude toward the importance of economic activities related to their forest. As shown in Table 3, timber sale was more important for forest owners who were unwilling to participate in the redemption, while forest owners who were willing to participate in the redemption were more convinced that they could obtain sustainable non-timber income from conservation projects.

According to our theoretical framework, the income of forest owners who participate in the redemption comprises the government's direct cash compensation (*D*) and the indirect non-timber income compensation incentives (*I*). If *D* is less than the sale obtained by logging commercial forests, the forest owners are encouraged to participate in the redemption program by continuously increasing their expected non-timber income in the future. Therefore, Timincome would have a negative impact on the possibility of forest owners participating in commercial forests' redemption, whereas *Nontimincome* would have a positive impact on this possibility.

The variable of owners' attitude toward the government governance (GG) included the owners' trust in government policies (Trust), the frequency of contact with forestry bureaus and other government departments due to disputes (Frequency), satisfaction with local ecology (Sat1), and satisfaction with ecological compensation of public welfare forest (Sat2). For comparison, as shown in Table 3, owners who were willing to participate had a higher degree of trust in government policies and were more satisfied with the local ecology and the ecological compensation of public welfare forests. Owners who were unwilling to participate had a higher frequency of contacts.

Trust had a positive impact on the willingness to participate. For forest owners, who have no bargaining power on the redemption price, the higher the owners' trust in government policies, the higher their acceptance of the redemption price. Furthermore, they would be more convinced that the government can fulfill its promises in the future. By contrast, the *Frequency* variable had a negative impact on their willingness to participate.

When disputes occur between different forest owners, between forest owners and the village collectives, or between forest owners and the government, the government's administrative efficiency in handling disputes was considered low, leading to higher transaction costs, and, thus, a decrease in forest owners' willingness to participate in the redemption. *Sat*1 and *Sat2* had positive impacts on their willingness to participate, as the more satisfied with the governance, the more willing the forest owners were to participate in environmental protection programs.

The factors that affect the willingness of forest owners to participate in the redemption are divided into *OH*, *PR*, *EI*, and *GG*. Accordingly, the model of the factors affecting the willingness of forest owners to participate is as follows:

$$W_i\ (y_i = 1 \mid X_i) = \alpha_0 + \alpha X_i + \mu_i \tag{3}$$

where $W_i$ represents the probability that forest owners are willing to participate in the redemption of commercial forests, $y_i = 1$ or 0, 1 means that forest owners are willing to participate in the redemption, and 0 means that they are unwilling to participate. $X_i$ is the independent variable that affects the willingness to participate in the redemption (Table 2), $\alpha_0$ is a constant term, and $\mu_i$ is a random error term.

The binary *probit* regression model was used to analyze the factors affecting the willingness to participate in the redemption. Under satisfying the basic assumptions, the probability of farmer *i*'s willingness to participate in the redemption of commercial forests is

$$Pr(y_i = 1) = \exp(\alpha X_i)/(1 + \exp(\alpha X_i)) \tag{4}$$

In Equation (4), Pr represents the probability of owner *i*'s willingness to participate in the redemption, X is a matrix containing independent variables, and $\alpha$ is the regression coefficient to be estimated.

## 5. Results and Discussion

The results of binary probit regression analysis are presented in Table 4. The log-likelihood value of the regression results was 30.35. The model showed a reasonable fit to the data, and all estimates had the expected values.

The results of the model revealed that the two variables of economic incentives (*EI*) (Timsale and Nontimincome) provided strong explanatory power for the likelihood of participation. The estimates provided some evidence that participation may be negatively correlated with the economic importance of timber sales (significant at the 1% level) and positively correlated with the owners' expectation of sustainable non-timber income from conservation programs (significant at the 1% level). These are consistent with the finding of Mitani and Lindhjem (2015).

To achieve improvement in social welfare through the redemption of commercial forests, the value of ecological services other than wood received by the public should be greater than the opportunity cost of the owners for redemption. The gain for the public is the consumer surplus (added value of ecological service minus the government's compensation), whereas the gain for the individual is the producer surplus (the compensation minus the opportunity cost). To achieve mutual benefit and win–win results, one feasible approach is to transfer the redeemed forests to state-owned forest farms and other entities for protection and management, which can benefit their own management experience and resources, in addition to drawing on the advantages of larger-scale management, including improved forests quality, healthy forest health, and ecotourism.

Forests have multiple uses, and some uses such as ecotourism and non-timber income are often not in conflict with forest conservation. Therefore, when direct cash compensation is not adequate to compensate for forest owners' losses, other alternative benefits are needed. These benefits include not merely monetary benefits but also employment or security benefits, especially for forest owners who rely heavily on forestlands; thus, only

increasing the redemption price will not fundamentally encourage them to participate in redemption.

**Table 4.** Probit estimation results.

| Dependent Variable | Model Participation Willingness (0/1) | | | |
|---|---|---|---|---|
| | **Coef.** | **S.E.** | **z** | ***p* > z** |
| Servant | 0.8085 | 0.8133 | 0.99 | 0.320 |
| Age | −0.0860 ** | 0.0382 | −2.25 | 0.024 |
| Edu | −1.1246 *** | 0.3804 | −2.96 | 0.003 |
| Income | 0.2127 ** | 0.0873 | 2.44 | 0.015 |
| Number | −0.6392 ** | 0.2910 | −2.20 | 0.028 |
| Size | 0.0347 ** | 0.0150 | 2.32 | 0.020 |
| Mature | 0.0248 *** | 0.0087 | 2.85 | 0.004 |
| Timsale | −0.6189 *** | 0.2318 | −2.67 | 0.008 |
| Nontimincome | 1.3243 *** | 0.4126 | 3.21 | 0.001 |
| Trust | 1.1526 *** | 0.3504 | 3.29 | 0.001 |
| Frequency | −2.0351 *** | 0.6717 | −3.03 | 0.002 |
| Sat1 | 0.6812 ** | 0.3479 | 1.96 | 0.050 |
| Sat2 | 0.2649 | 0.2772 | 0.96 | 0.339 |
| Constant | 0.2225 | 3.0750 | 0.07 | 0.942 |
| Number of obs. | 218 | | | |
| Log-likelihood | −30.35 | | | |
| Pseudo-$R^2$ | 0.51 | | | |

** $p < 0.05$; *** $p < 0.01$.

The estimation results of the model revealed that the three government governance (*GG*) variables (Trust, Frequency, and Sat1), but not Sat2, provided strong explanatory power for the likelihood of participation. We found statistically significant evidence at the level of 1% that the owner's trust in government policies increases the likelihood of participation in redemption. The direct cash compensation is determined by the local government. Whether forest owners can obtain continuous non-timber income in the future is also related to whether the government can transform the leading forestry industry. Therefore, higher trust in government policies increases willingness to participate in enrollment.

The results also suggested that the probability of participation was negatively correlated with the frequency of contact with forestry bureaus and other government departments due to disputes (significant at the 1% level). With the increase in the frequency of disputes, the government's administrative efficiency is lower, which increases the transaction costs of the redemption. Therefore, the willingness of forest owners to participate in the redemption was reduced.

Satisfaction with the local ecology was found to be statistically significant at 5%, thus increasing the likelihood of participation. Local forest owners who also understood the importance of environmental improvement were more willing to contribute to the ecology of their hometowns and were willing to bear part of the cost of conservation. Although some forest owners received direct cash compensation lower than their expectations, they also participated in the program. Forest owners realize that the implementation of environmental protection policies can not only achieve a variety of ecological and environmental benefits but also additional social and economic benefits [18,19].

The probability of participation was found to be negatively correlated with the owners' age (significant at the 5% level), education (significant at the 1% level), and the number of elderly people (significant at the 5% level). The older the owner, the fewer opportunities, the greater the dependence on forest land resources, and the less willingness to participate in the redemption. The same rationale also applied to the number of elderly people in the household. The negative relationship with education is consistent with the results of Cai and Tan (2020), who claimed that more educated owners have more interest and enthusiasm to manage forests, leading to their less willingness to participate in conservation.

The owners' income, the size of the owned forest, and the estimated proportion of mature forest were found to have statistically positive correlations with participation. A higher income may put less pressure on the owner and reduce their dependence on income from wood products. It was found that the owners with larger forest sizes and larger proportions of mature forests were more interested in the program, as they had invested more in silviculture and management in the past. Compensation by enrollment in the program may reduce the potential risk resulting from restrictions in the utilization of the forests.

## 6. Concluding Remarks and Policy Implications

Our analyses and results suggested that economic incentives and better governance motivated the owners to participate in the redemption of commercial forests. While direct cash compensation determined by the government's maximum price limit was important, indirect compensation for non-timber income also played an important role to make up for the loss of timber sales.

More trustful government and its policies, higher government efficiency, and satisfaction with the local ecological governance were found to promote the household's engagement in the program of redemption.

Policymakers may utilize the final offer arbitration method to improve the redemption price, that is, direct cash compensation incentives. The proposed method is like the widely practiced arbitration method; specifically, both parties bid at the same time and then notify the arbitration department, which chooses one of the two as the final price [20]. For the redemption of commercial forests, the local government entrusts a forest resource asset appraisal company to carry out the appraisal under the approval of forest owners. When the appraisal price is lower than the maximum limit price, the redemption price is the appraisal price, and when the appraisal price is higher than the maximum limit price, the redemption price is the maximum limit price. From a comparison of the maximum limit price with the appraisal price, it is evident that this can become a third-party game between the government, the owner, and the resource asset appraisal companies. The government provides the maximum price (*WTP*), the owner provides the lowest acceptable price (*WTA*), and the company provides the appraisal price. The appraisal price is the arbitration price, and whichever is closer to the arbitration price determines the redemption price.

Transparent and effective information can reduce the cost of redemption transactions. Forest owners' trust in the government stems from fair and transparent decision making and timely fulfillment of promises. Individuals can only act based on the information they have. Therefore, the government can transmit information at a lower cost, encourage the parties to report information truthfully, and act in accordance with preset rules to achieve the established social goals the government wants to achieve. Further, it becomes important for policymakers to consider campaigns and program designs that can increase forest owners' expectations of sustainable non-timber income and deliver on such expectations. Thus, it is necessary to establish and improve a forest rights trading center and platform, which can reduce transaction costs such as information search, forest resource asset assessment, redemption negotiation, and redemption contract execution, besides ensuring the credibility of forest rights transactions.

**Funding:** This research was funded by the National Social Science Foundation of China, grant number 19XGL013, funded by the National Scholarship Fund.

**Institutional Review Board Statement:** Not applicable.

**Informed Consent Statement:** Not applicable.

**Data Availability Statement:** Not applicable.

**Acknowledgments:** I am thankful for the financial support of the National Social Science Foundation of China and the National Scholarship Fund. I express my appreciation to the anonymous referees and editors of the journal for their constructive comments and suggestions.

**Conflicts of Interest:** The authors declare no conflict of interest.

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
