# Peer review of "Households’ Willingness to Accept Forest Conservation and Ecosystem Services"

_forests, doi:10.3390/f13091399_

Round 1
Reviewer 1 Report
The topic is timely and important. While the methods may have been sound, the authors need to do a better job at explaining how the data collection was conducted. This paper needs extensive proofreading. Many parts are not clear because of this.

Reviewer 2 Report
Thank you very much for reading this text. The text is certainly valuable, but it requires minor additions and clarification of individual threads.
The article is methodologically and technically well organized. The research is exploratory. The author discusses an important topic of the readiness of households to accept forest and ecosystem protection services
The introduction presents an outline of the topic and an attempt to examine the basic factors in dealing with the conflict between logging and the economic use and protection of forests within the state budget.
240 households were surveyed, generating 218 valid questionnaires. Tables 1 and 2 present the variables used for the empirical analysis, their definition, as well as the mean and standard deviation. The dependent variable (Willingness) is the willingness of the forest owner to participate or not participate in the remission. The statistical tests are described in detail and are summarized in the appropriate tables
One aspect that could be a bit more specific is the reference to studies by other authors in the review, but I understand that these aspects could constitute a separate work. It is suggested that the presentation of tables in 1 and 2 should be considered / taken into account as more readable.
A detailed discussion takes up the strengths and needs, and indicates the limitations of the research. The article also contains indications for the future, which is a very valuable remark for future researchers of this topic. The small number of published studies on this subject shows that comprehensive research is needed in this area.
